# Green Synthesized Chitosan/Chitosan Nanoforms/Nanocomposites for Drug Delivery Applications

**DOI:** 10.3390/polym13142256

**Published:** 2021-07-09

**Authors:** Iyyakkannu Sivanesan, Judy Gopal, Manikandan Muthu, Juhyun Shin, Selvaraj Mari, Jaewook Oh

**Affiliations:** 1Department of Bioresources and Food Science, Konkuk University, Hwayang-dong, Gwangjin-gu, Seoul 05029, Korea; isivanesan@gmail.com; 2Laboratory of Neo Natural Farming, Chunnampet, Tamil Nadu 603 401, India; jejudy777@gmail.com (J.G.); bhagatmani@gmail.com (M.M.); 3Department of Stem Cell and Regenerative Biotechnology, Konkuk University, Seoul 143-701, Korea; junejhs@konkuk.ac.kr; 4Department of Chemistry, Guru Nanak College, Chennai 600 042, India; mariselvaraj@yahoo.com

**Keywords:** chitosan, chitosan nanoparticles, synthesis, green synthesis, drug delivery

## Abstract

Chitosan has become a highlighted polymer, gaining paramount importance and research attention. The fact that this valuable polymer can be extracted from food industry-generated shell waste gives it immense value. Chitosan, owing to its biological and physicochemical properties, has become an attractive option for biomedical applications. This review briefly runs through the various methods involved in the preparation of chitosan and chitosan nanoforms. For the first time, we consolidate the available scattered reports on the various attempts towards greens synthesis of chitosan, chitosan nanomaterials, and chitosan nanocomposites. The drug delivery applications of chitosan and its nanoforms have been reviewed. This review points to the lack of systematic research in the area of green synthesis of chitosan. Researchers have been concentrating more on recovering chitosan from marine shell waste through chemical and synthetic processes that generate toxic wastes, rather than working on eco-friendly green processes—this is projected in this review. This review draws the attention of researchers to turn to novel and innovative green processes. More so, there are scarce reports on the application of green synthesized chitosan nanoforms and nanocomposites towards drug delivery applications. This is another area that deserves research focus. These have been speculated and highlighted as future perspectives in this review.

## 1. Introduction

Chitosan, a natural polysaccharide is second after cellulose in terms of abundance, usage, and distribution [1]. Chemically, chitosan consists of glucosamine and *N*-acetyl glucosamine residues [2]. Chitosan [(1,4)-2-amino-2-deoxy-D-glucan] is a linear polyaminosaccharide obtained subsequent to *N*-deacetylation of chitin. Chitin is the structural component of the exoskeleton of shrimps, lobsters, and crabs; it is also present in the cell walls of fungi and yeast, squid pens [3], green algae, and cuticles of insects and arachnids. Chitosan has gained great impetus owing to its biological properties and applications in the medical, food, and agricultural sectors [4,5,6]. The recovery of chitosan from marine shell waste and from marine food waste generated by food-processing industries makes this polymer one of the most important renewable assets.

Chitosan’s therapeutic properties, such as inhibition of microorganisms, pain alleviation [7,8], promotion of hemostasis, and epidermal cell growth [9] are rather unique. Much interest is targetted towards their potential applications in medical and pharmaceutical roles. The increased interest in chitosan is attributed to its favorable properties, such as biocompatibility, ability to bind some organic compounds, susceptibility to enzymatic hydrolysis, and intrinsic physiological activity combined with nontoxicity [10,11,12]. These properties come handy for biomedical applications such as drug delivery and targeting, wound healing and tissue engineering, as well as in nanobiotechnology. Chitosan has, because of its biological and physicochemical properties, become an attractive option as a drug delivery element and for delivery of other macromolecules [13,14,15]. Chitosan-based delivery systems range from microparticles to nanoparticles (NPs), nanoparticle (NP) composites, nanofibres, and films. Chitosan combines with nanometals such as iron (Fe), copper (Cu), silver (Ag), silicon (Si), zinc (Zn), zinc oxide (ZnO), and titanium dioxide (TiO_2_) to form nanocomposites, leading to enhanced anti-microbial properties [16,17,18,19,20,21].

The advent of nanotechnology and introduction of NPs has resulted in their diverse applications [22]: biomedical, drug delivery, antitumour activity, tissue engineering, sensor development, pathogen detection, protein detection, gene delivery, environmental remediation and water purification, and various similar applications [23,24,25,26,27,28,29]. NPs have become very attractive options for diagnosis and therapeutics, owing to their unique properties. Small sizes with large surface area to volume ratio, as well as their stability and enhanced translocation into the cells, are handy assets equipping these NPs for diverse applications. NPs exhibit exceptional optical properties, rendering them suitable for imaging applications (based on their ability to produce quantum effects) [27,28,29]. The most commonly studied metallic NPs include gold (Au), Ag, aluminum (Al), Zn, Fe, and TiO_2_ NPs [30]. Researchers in an attempt to enhance the properties of these metallic NPs have conjugated/modified/functionalized these with various biomolecules and ligands [30,31,32].

Chitosan NPs (Ch NPs) have been recently gaining a lot of popularity. Its bio nature, abundance, degradability, and special properties are its strengths. Chitosan can be easily processed in diverse forms, such as films, threads, tablets, membranes, and microparticles/nanoparticles. This allows to design a variety of medical and pharmacological devices adaptable to end purposes. Ch NPs are reported to exhibit antitumor properties by improving the body’s immune function [33,34,35,36,37,38]. Chitosan is useful in bandages to reduce bleeding and as an antibacterial agent and for drug delivery.

NPs synthesis requires the use of reactive and toxic reducing agents, which may result in death and adversely affect the surrounding environment and organisms [39]. Green synthesis is an alternative approach. The green synthesis method has an edge over chemical and physical methods because it is cost-effective, is environment friendly, and there is no requirement for high pressure, energy, temperature, and toxic chemicals [40]. Furthermore, the synthesis of NPs using biological resources is even more attractive, since it is rapid, eco-friendly, and non-toxic. Biological material includes microorganisms such as bacteria [41] and fungus [42], algae [43,44], while plants have also become an alternative option. Nowadays, plant-mediated synthesis is preferable compared to microbe-mediated synthesis due to its simplicity, rapidity, and avoidance of cumbersome culture maintenance processes [45,46]. Plant materials like leaves [45,47], flowers [48,49], seeds [50,51], stems [52,53], fruits [54,55,56], peels [57,58], and weeds [59] have been used for the synthesis of NPs.

In the following paper, we review the various methods for recovery of chitosan from chitin. The methods used to prepare chitosan nanoparticles and nanoforms are also presented briefly. The current status on the available techniques for green synthesis of chitosan/nanochitosan has been reviewed for the first time. The drug delivery applications of chitosan and nanochitosan have been discussed and the need for introducing green synthesized chitosan nanomaterial for drug delivery options has been proposed.

## 2. Chitosan/Chitosan NPs Synthesis

### 2.1. Preparation of Chitosan

Chitosan is not generally available in nature; it is recovered from chitin, a naturally available polymer. Mucor, Absidia, and Rhizopus species, that are representative members of Mucorales are the only exceptions, where the polymer chitosan naturally exists [60]. Chitin, in turn, is extracted from discarded remnants of shrimp, squid, prawn, lobster, crab, and domestic marine shell wastes. Crustacean shells are composed of about 15–40% chitin and 20–40% protein, 20–50% calcium and magnesium carbonate and astaxanthin, lipids, and other minerals in trace quantities [61]. Isolation techniques are diverse because the sources are diverse and their composition is variable [62]. A chemical process is mostly initiated for extracting the protein and removing inorganic matter. A bleaching step is optional, using a solvent or through oxidation of pigments [63]; this is followed by demineralization of shells using dilute HCl/HNO_3_/H_2_SO_4_/CH_3_COOH [64]. Deacetylation of chitin by hydrolysis of the acetamide groups with concentrated NaOH or KOH at temperatures above 100 °C yields chitosan. The degree of acetylation (DA) of chitosan depends on the deacetylation conditions [65]. Lertwattanaseri et al. report a microwave technique to obtain chitin whiskers [66].

Alternative treatments use ethylene diamine tetra acetic acid (EDTA) [67] or ionic liquid extraction [68,69]. Lactic acid fermentation has been reported for the extraction of chitin from prawn shells [70]. Enzymatic extracts or isolated enzymes and microbiological fermentation have been tested [71], but is time-consuming and results in 1–7% of residual protein [72]. Other biotechnological processes that use enzymatic deacetylation of chitin have been demonstrated as alternatives for chemical processing. Chitin deacetylases have been used to hydrolyze *N*-acetamide bonds, resulting in chitosan [73]. These enzymes are obtained from some select fungi and insects. The enzymatic deacetylation process is reported to enhance the degree of acetylation and average molecular mass of chitosan. However, this alternative is still in the lab [62,74].

### 2.2. Preparation of Ch NPs

Ch NPs have been reported to be prepared by emulsion droplet coalescence [75], a reverse micellar method, ionic gelation [76,77], precipitation [78], sieving, and spray drying [79]. All of the above-described techniques follow a bottom-up approach. Bottom–up techniques arrange smaller components into complex assemblies and top-down approaches begin with large sized materials and break them into smaller ones. Routine conventional NPs synthesis usually follow bottom–up techniques.

Chitosan micro- and nanoparticles have been prepared using varied techniques. The particle size, stability of the active constituent and the final product, residual toxicity present in the final product, and their drug release kinetics are what go into the selection of an appropriate preparation method [78]. It is confirmed that the size of the prepared particles depends on the molecular weight and chemical structure and degree of deacetylation (DDA) of chitosan, including the method used. The higher the molecular weight of chitosan, the larger the particle size [80,81]. The most common methods for obtaining Ch NPs are: ionotropic gelation, microemulsion, emulsification solvent diffusion, and emulsion-based solvent evaporation. Each of these methods influence the particle size and surface charge of nanochitosan and impact the molecular weight and degree of acetylation. The coacervation method involves the separation of spherical particles by mixing electrostatically driven liquids [82,83]. In the polyelectrolyte complex (PEC) method, an anionic solution is added to the cationic polymer, under mechanical stirring, to obtain nanoparticles [84,85]. The coprecipitation method involves the addition of a chitosan in low pH to a high pH solution, resulting in coprecipitation of highly monodisperses chitosan nanoparticles [86]. In the microemulsion method, chitosan in acetic acid solution and glutaraldehyde are added to a surfactant in an organic solvent such as hexane. NPs form overnight as the cross-linking process is completed, resulting in the formation of small-sized nanoparticles [87]. The Emulsification Solvent Diffusion Method is where an o/w emulsion is prepared with mechanical stirring and high pressure homogenization [88,89] to achieve 300–500 nm sized Ch NPs. Emulsion Based Solvent Evaporation Method is a slight modification of the above method but avoids high shear forces. In reverse micellar method, the surfactant is dissolved in an organic solvent, to which chitosan, and drug and crosslinking agents are added under constant overnight vortex mixing, leading to the formation of Ch NPs of fine sizes [90].

## 3. Green Synthesis of Chitin/Chitosan/Chitin and Chitosan NPs

Chemical extraction of chitosan has its own drawbacks: (i) the physico-chemical properties of chitin are affected and MW and DA decrease, negatively affecting intrinsic properties; (ii) wastewater effluents contain some chemicals, and (iii) increased cost of purification processes. This is why biological/green extraction techniques are gaining popularity. Biological synthesis uses enzymes and microorganisms for chitin extraction and chitosan recovery. Usually, as in the case of all nanomaterial synthesis, two distinct categories are nominated—chemical synthesis or green synthesis. Here, in the case of chitin/chitosan-based materials, chemical synthesis was the pioneering technique; biological methods of chitin and chitosan preparation then gained popularity. With the existence of a middle term, already established by numerous publications, green synthesis as such in the case of chitin research has become hard to clearly demarcate. As supporters of the fact that biological methods are indeed green synthesis methods, we categorize biological methods of preparation within the topic of green synthesis.

### 3.1. Biological Method

Khanafari et al. [91] compared chemical versus biological extraction of chitin from shrimp shells. The biological method (using microorganisms) was demonstrated to be better than the chemical method, since the structural integrity of chitin was preserved. Bustos and Healy [92] confirmed that chitin obtained by the deproteinization of shrimp shells with various proteolytic microorganisms has higher molecular weight. The biological extraction of chitin has the following advantages: high reproducibility in shorter time, simple manipulation, less solvent consumption, and lower energy input. However, the biological method is still limited to laboratory scale studies. Recently, two reports have elaborately reviewed the most common biological methods used for chitin extraction [71,93]. The lactic acid bacterial fermentation process has been studied more extensively by Guerrero Legarreta et al. [94] and Cira et al. [95]. Enzymatic deproteinization of chitin requires the use of proteases. Proteolytic enzymes such as alcalase, pepsin, papain, pancreatine, devolvase, and trypsin are mainly obtained from plant, microbe, and animal sources; these are involved in deproteinization of crustacean shells. Alcalase 2.4 L (Novo Nordisk A/S) is a serine endopeptidase obtained from *Bacillus licheniformis*; this has been used for the isolation of chitin containing about 4% protein impurities [72]. Such purity is sufficient for many non-medical applications of chitin [96]. Manni et al. [97] reported the isolation of chitin from shrimp waste using *Bacillus cereus* SV1 crude alkaline proteases. In another study, enzymatic deproteinization was optimized by Younes et al. [98]. In this study six alkaline crude microbial proteases from *Bacillus mojavensis*, *Bacillus subtilis*, *B. licheniformis*, *B. licheniformis*, *Vibrio metschnikovii,* and *Aspergillus clavatus* were used. Mukhin and Novikov [99] used crude proteases isolated from the hepatopancreas of crab. Younes et al. [62] used alkaline proteases from the red scorpionfish *Scorpaena scrofa* [100]. Here, the excessive cost of using enzymes can be decreased by performing deproteinization using a fermentation process, using an endogeneous microorganisms (auto-fermentation) as fermenter or by adding selected strains of microorganisms [71]. Fermentation methods could be via lactic acid fermentation or non-lactic acid fermentation. (a) Lactic acid fermentation of crustacean shells utilizes *Lactobacillus* sp. that produce lactic acid and proteases. (b) In non-lactic acid fermentation, both bacteria and fungi were used: *Bacillus* sp. [101,102,103], *Pseudomonas* sp. [70,104,105], and *Aspergillus* sp. [106].

Sini et al. [102] studied the fermentation of shrimp shells in jaggery broth using *B. subtilis* [104,107,108]. Ghorbel-Bellaaj et al. [109] elaborately studied the fermentation efficiency of *P. aeruginosa*. Teng et al. [110] evaluated the production of chitin from shrimp shells and fungi in a one-pot fermentation process, where fungal proteases hydrolyze proteins into amino acids. Recently, Younes and Rinaudo extensively reviewed the preparation methods involved in the recovery of chitin and chitosan from marine sources [98]. Their review has dealt with the state-of-the-art methodologies involved in the recovery of chitin/chitosan and their future perspectives.

### 3.2. Green Synthesis for Chitin/Chitosan/Chitin NF/Ch NPs and Ch NF

Actually there are quite a number of studies that describe the use of chitin/chitosan in the green synthesis of inorganic metal NPs. They have also been used as stabilizers during NPs synthesis. However, the term ‘green synthesis’ of chitin/chitosan/Ch NPs’ did not show many results. Green synthesis of chitin and its associates, as we reviewed, was observed to be only supported by scattered scanty reports. We use the term ‘scattered’ because these reports are single publications on this topic, which have not been further researched or developed on. Here, we present the scattered information available on this subject.

Recently, Madalloni et al., 2020, reviewed the green synthesis of chitin using deep Eutectic Solvents (DESs) and Natural Deep Eutectic Solvents (NADESs) [111,112,113]. In this extraction procedure, shrimp shells are treated with 10% citric acid (this is an edible, weak acid that can be extracted from natural sources, used instead of HCl), for demineralization [114]. Deproteinization includes using microwave irradiation, on the above-mentioned pretreated samples suspended in different DES solutions. Finally, simple centrifugation allows for the separation of chitin from DES. Another group [115] developed a zero-waste method to convert shrimp shell waste into chitin in NADESs rather than in DESs. Liu et al. [116] reported an efficient and green chemical process using glycerol to reduce the NaOH content that is usually utilized. Glycerol is a recyclable, stable, green solvent that can be obtained as a by-product of biodiesel. The chitin to chitosan reaction involves the treatment of chitin with 30% NaOH and glycerol, keeping a 1:40 chitin/glycerol ratio, with water being the only other additive. The superiority of this method is that both glycerol and NaOH can be recovered and reused again.

Green synthesis of Ch NPs is also represented by just a few reports. Ch NPs are generated by ionic cross-linking between chitosan and sodium tri-polyphosphate (TPP); there is a report where antimicrobial Ch NPs were synthesized by chemical cross-linking with cinnamaldehyde, another eco-friendly bactericidal agent [117]. Bacterial leaf blight caused by *Xanthomonas oryzae* devastates rice crops. The antibacterial activity of biosynthesized Ch NPs against this rice pathogen has been reported [118].

A pair of grinding stones can effectively disintegrate chitin organization in crab shells. Mechanical grinding is another simple, yet powerful method allowing the recovery of chitin NFs from waste crab shell in large amounts [119]. The chitin consisted of highly uniform NFs with a width of approximately 10 nm. The same Ifuku group also reported the recovery of chitin NFs from shrimp shells, mushrooms, and squid pens [120,121,122]. Grinding does not involve any complicated chemicals and hence is an eco-friendly process [123]. Few reports of chitin nanofiber synthesis have also been noted with diameters of 3 nm being fabricated in hexafluoroisopropanol (HFIP) through a self-assembly strategy [124,125]. Although chitin NFs have been prepared in HFIP using self-assembly, HFIP is toxic. “Green” solvents such as ionic liquids and urea–NaOH mixtures are preferred in place of HFIP. Qin et al. successfully reported the use of environmentally friendly ionic liquids to obtain high molecular weight purified chitin and chitin films and nanofibers [126,127,128]. Chitosan nanofibers (Ch NFs) have also been synthesized via electrospinning [129]. Polyethylene oxide and poly (vinyl alcohol) (PVA) are often used to blend with chitosan solutions [130,131]. The authors of [132] report a simple and green method for the preparation of chitosan and chitosan-based nanofibers by freeze-drying dilute aqueous solutions, without the use of organic solvents, high concentration of acid solutions, or the need to pre-treat chitosan. Ch NFs with diameters ranging from 100 to 700 nm were obtained from aqueous chitosan solutions. Chitosan/PVA blend NFs with different mass ratios were produced by freeze-drying [130,131].

Our research group [133] recently reported an eco-friendly, sustainable phytomediated one pot recovery of chitosan from commercial chitin and from crab and shrimp shells and squid pen solid wastes. Graviola extracts were employed for the recovery of Ch NFs. Graviola contains acetogenins that actively interact with chitin in insects. With that as the core idea, the graviola extracts were chosen for orchestrating chitin recovery and a possible chitin to chitosan transformation mediated under magnetic stirring on a hot plate.

Silver nanoparticles (AgNPs), particularly those entrapped in polymeric nanosystems, have arisen as options for managing plant bacterial diseases. Chitosan, owing to its low cost, good biocompatibility, antimicrobial properties, and biodegradability, is the polymer under high consideration. Authors have reported green-synthesized Ag NPs using different concentrations of aqueous extract of tomato leaves, followed by entrapment of AgNPs with chitosan (Ch-AgNPs). They used green synthesized systems for controlling tomato bacterial wilt caused by *Ralstonia solanacearum* [134]. Green synthesis procedures have also been demonstrated for the synthesis of chitosan bionanocomposites. Silver-based chitosan bionanocomposites have been synthesized using the stem extract of *Saccharum officinarum* [135]. The antibacterial activity of these silver-based chitosan bionanocomposites was evaluated against *Bacillus subtilis*, *Klebsiella planticola*, *Streptococcus faecalis*, *Pseudomonas aeruginosa*, and *Escherichia coli*. Kim et al. [136] used a green route to produce Au NPs in a chitosan matrix, whose functional groups favor the interaction with caffeic acid. Shameli et al. [137] demonstrated the green synthesis of Ag/montmorillonite (MMT)/chitosan bionanocomposites using the UV irradiation method and demonstrated their antibacterial activity against Gram-positive and Gram-negative bacteria. Ch NFs with silica phase (Ch NFs/silica) were synthesized by an electrospinning technique to obtain highly porous 3D NF scaffolds. Silver nanoparticles in the form of a well-dispersed metallic phase were synthesized in an external preparation step and embedded in the Ch NFs /silica to form Ag/Ch/silica nanocomposites [138]. Green synthesis of Ag-Ch nanocomposites using chitosan as a reducing agent as well as a stabilizing agent and NaOH as accelerator is reported [139]. The Ag/Ch nanocomposite gel is transformed into colloid by dissolving into chitosan solution and its antibacterial activity against *E. coli* and *S. aureus* bacteria was demonstrated. In another work [140], colloidal Au NPs were stabilized into a chitosan matrix and prepared using a green route. The use of chitosan, with a large number of amino and hydroxyl functional groups, enables the simultaneous synthesis and surface modification of AuNPs in one pot. These hybrid nanocomposite films were used as sensors for the determination of caffeic acid, an antioxidant that has recently attracted much attention because of its benefits to human health. A novel Ch-Fe_2_O_3_ nanocomposite was synthesized by a facile one pot green route [141]. The nanocomposite showed excellent recyclable efficiency up to five cycles. The nanocomposite has also been proved to be an excellent potential sorbent for recovery of toxic elements from industrial and medical wastewater. Using cinnamaldehyde (CA), a natural preservative, researchers have prepared CA/SA/Ch NPs nanocomposites. In vitro release experiments showed that CA/SA/Ch NPs had the function of sustained release, indicating that SA/Ch NPs can be used as a promising carrier for CA [142]. Antimicrobial carboxymethyl chitosan-nanosilver (CMC-Ag) hybrids with controlled silver release was fabricated by Huang et al. [143]. Under principles of green chemistry, the synthesis was conducted in an aqueous medium exposed to microwave irradiation for 10 min with non-toxic chemicals. Their antibacterial activity was tested against *Staphylococcus aureus* and *Escherichia coli*.

Ultrasound-induced synthesis of Ch-modified nano-scale graphene oxide (Ch-NGO) hybrid nanosheets, which has great potential pharmaceutical applications, in supercritical CO_2_ without catalyst is reported [144]. Ciprofloxacin hydrochloride (CIP) was incorporated into a green based nano-composite (HS/Ch-NC) to control the antibiotic release and increase its bioavailability [145]. An in vitro drug release study of the nano-composite HS/Ch-NC showed high activity against gram negative bacterial strains due to the successful release of CIP from the chitosan composite into the tested bacterial strains. Chitosan nanoparticles (NPs) are widely studied as vehicles for drug, protein, and gene delivery. However, lack of sufficient stability, particularly under physiological conditions, render chitosan NPs limited pharmaceutical utility. Stable Ch NPs suitable for drug delivery applications [146] were prepared by grafting to phthalic or phenylsuccinic acids. Subsequently, polyphosphoric acid (PPA), hexametaphosphate (HMP), or tripolyphosphate (TPP) were used to achieve tandem ionotropic/covalently crosslinked chitosan NPs in the presence of 1-ethyl-3-(3-dimethylaminopropyl)-carbodiimide (EDC). Chitosan-zinc oxide nanoparticle composite [147] and CuO/Chitosan composites [148] are also reported. Table 1 presents a consolidated list of chitin family of green synthesized materials.

## 4. Drug Delivery Applications of Chitosan/Nanochitosan

### 4.1. Basics of Polymers in Drug Delivery

Nanomaterials or nanoparticles offer new opportunities in material science and biomedicine. The small size of nanoparticles allows them to enter cells and organelles for targeted drug delivery [149]. NPs can be conjugated with ligands or antibodies to enable recognition and binding to specific receptors on cell targets [150,151]. Different methods are available to prepare chitosan micro-/nanoparticles in which the drug is mostly bound to chitosan by hydrogen bonding, electrostatic interaction, or hydrophobic linkage. There are several mechanisms which govern drug release from chitosan nanoparticles, such as swelling of the polymer [152], diffusion of the adsorbed drug, drug diffusion through the polymeric matrix, polymer erosion or degradation, and a combination of both erosion and degradation [147], as represented in Figure 1. The swelling of the polymer is characterized by the imbibition of water into the polymer until the polymer dissolves. This drug release mechanism is characterized by the solubility of the polymer in water, or the surrounding biological medium [153]. Erosion and degradation of polymers are interrelated features. Sometimes, degradation of the polymer may cause subsequent physical erosion as bonds break. Erosion of polymers is a complex phenomenon as it involves swelling, diffusion, and dissolution.

### 4.2. Chitosan Microparticles-Based Drug Delivery

Among the novel drug delivery systems investigated, chitosan micro-/NPs offer great promise in oral, parenteral, topical, and nasal applications. In these systems, the drug is either confined and surrounded by a polymeric membrane or is uniformly dispersed in the polymer matrix. Drug release at a specific site and for an extended period of time could also be achieved by mucoadhesion, where chitosan adheres to specific mucosal surfaces in the body, such as buccal, nasal, and vaginal cavities [154,155,156]. Chitosan microparticles have shown varied applications in the delivery of a range of compounds owing to particle size reduction by micronization and their mucoadhesive properties. Dastan and Turan [157] developed chitosan–DNA microparticles and reported a sustained-release profile of DNA, with a potential transfer of the DNA into human embryonic kidney, Swiss 3T3, and HeLa cell lines. Another research group prepared chitosan–DNA microparticles for mucosal vaccination in simulated intestinal fluid and simulated gastric fluid [158]. Luteinizing hormone-releasing chitosan-based microparticles as a vaccine delivery vehicle are also reported. Successful delivery of hormones by these particles extends their application for induction of immunity against some tumor antigens and microorganisms such as bacteria and viruses [159].

Insulin delivery via the nasal route using chitosan microparticles was demonstrated by Varshosaz et al. [160]. They demonstrated that insulin-loaded microspheres exhibited a 67% lowering in blood glucose level, compared to insulin administered intravenously. Many research groups have thus described the applicability of chitosan microparticles in drug delivery. The encapsulation of diclofenac sodium [161], 5-flurouracil [162], cisplatin [163], felodipine [164], and hydroquinone [165] into these carriers has been reported, and the designed microparticles generally exhibit a controlled-release effect. Chitosan magnetic microparticles (CMM) are a special class of chitosan microparticles that have been developed and used for the delivery of anticancer drugs or radionuclide atoms to a targeted tissue [166] by binding the drug or the radioactive atom to a magnetic compound, which is then injected into the blood and stopped at the targeted tissue by an externally applied magnetic field [167,168]. Attapulgite, a nanosized silicate clay naturally occurring polymer used in drug delivery [169], has been introduced into cross-linked diclofenac sodium chitosan microspheres in which the prepared chitosan/attapulgite hybrid microspheres exhibited narrow size distribution and minimum drug release in the simulated gastric fluid [170].

### 4.3. Ch NPs-Based Drug Delivery

The preparation of curcumin-loaded Ch NPs has been reported to enhance the drug solubility and stability in the gastro intestinal tract [171]. Facilitation of the transmucosal delivery of two hydrophobic drugs, triclosan and furosemide, has been achieved by developing drug-loaded Ch NPs [172]. Low-molecular weight heparin (LMWH) has been loaded into Ch NPs and showed improved oral absorption and relative bioavailability, compared to a solution of LMWH [173]. Recently, chitosan nanotherapeutics have received great attention in the field of oncology because of their enhanced tumor targeting, ability to load different hydrophobic anticancer drugs, and the ability to control anticancer drug release rates [174,175]. Chitosan-loaded paclitaxel NPs exhibited excellent tumor-homing ability in tumor-bearing mice [176,177,178,179,180]. Protein/siRNA-loaded Ch NPs have been prepared and have shown 98% entrapment efficiency with adequate stability [164]. BSA has been encapsulated into Ch NPs [181]. Genetic immunization using Ch NPs-loaded plasmid DNA was investigated, and the results showed measurable and quantifiable levels of gene expression and considerable antigen titer [182]. Other workers investigated the potential of these nanoparticles as carriers for antigens by using recombinant hepatitis B surface antigen [183]. Ch NPs for oral delivery of insulin have been successfully formulated and demonstrated for enhanced in vitro as well as in vivo absorption and improved insulin bioavailability. [184] Magnetic nanosized Ch NPs were also developed as a special type of Ch NPs and showed sustained drug release with minimal toxicity. Camptothecin magnetic Ch NPs in which polyethylene glycol was coupled with magnetic nanoparticles to increase their biocompatibility has been reported [185].

Ch NPs have been successfully used for drug delivery. Mohammed et al. elaborately reviewed the extensive milestones achieved through Ch NPs-based drug delivery [186]. The effective delivery of catechin and epigallocatechin across interstinal membranes was achieved by encapsulation within Ch NPs [187]. Permeation of tamoxifen across intestinal epithelium was enabled via formulating tamoxifen into lecithin-chitosan NPs [188]. Feng et al. [189] reported successful oral delivery of anticancer drugs, using nanoparticles of doxorubicin hydroxide (DOX)/chitosan/ carboxymethyl chitosan formulations, which could enable enhanced absorption of DOX throughout the intestine. Alendronate sodium is used in the treatment of osteoporosis, and alendronate sodium was encapsulated in Ch NPs. This overcame the low oral bioavailability and gastrointestinal side-effects that this drug confronted during drug delivery [190]. Sustained delivery of sunitinib, a tyrosine kinase inhibitor, up to 72 h, was achieved though encapsulation of Ch NPs [191]. Insulin--loaded Ch NPs crosslinked with TPP increased their uptake by the intestinal epithelium [192]. Bay 41-4109, an active inhibitor of hepatitis B virus, was formulated as Ch NPs to improve drug solubility and oral bioavailability [193]. Chitosan and carboxymethyl Ch NPs were found to be excellent carriers for oral vaccine delivery of extracellular products of *V. anguillarum* (pathogenic bacteria). This chitosan-based formulation increased its stability, resulted in sustained release, and protected the antigenic protein from entering the spleen and the kidney, which is critical for immune response [194].

Chitosan is biodegradable, biocompatible, exhibits low toxicity, adheres to mucus, and opens the tight junctions of the nasal membrane. Owing to these properties, chitosan has applications in nasal delivery [195]. Carboxymethyl Ch NPs of carbamazepine (treats epilepsy) enhanced the bioavailability and brain targeting via the nasal route. The bioavailability of leuprolide (used to treat prostate cancer and hormone-dependent diseases), increased when formulated as thiolated-Ch NPs [196,197]. Islam and Ferro [198] reviewed the various modes of Ch NPs-based nanoparticle-aided drug delivery to the lungs. The authors claimed that the positive charge on the surface of chitosan provides mucoadhesive properties, increasing the potential for drug absorption. In addition, the positively charged chitosan are able to open the intercellular tight junctions of the lung epithelium, increasing uptake. More so, chitosan binds to phosphoryl groups and lipopolysaccharides on bacterial cell membranes, and helps in fighting pulmonary bacterial infections. A dry powder inhalation (DPI) of rifampicin, an antitubercular drug, was formulated with chitosan, bringing about sustained drug release until 24 h and no toxicity to cells nor organs [199]. The antifungal drug Itraconazole, which is used to treat pulmonary infections, suffers from low solubility. Its aerosolization properties, which would enable its pulmonary deposition, was improved by formulating it with spray dried chitosan NPs with lactose, mannitol, and leucine [200].

Giovino and co-workers have investigated chitosan buccal films of insulin-loaded poly (ethylene glycol) methyl ether-block-polylactide (PEG-b-PLA) NP [201]. Polysaccharide-based NP chitosan [202] and curcumin prepared as polycaprolactone nanoparticles coated with chitosan and NP encapsulation of enriched flavonoid fraction (EFF-Cg) obtained from *Cecropia glaziovii* were successfully demonstrated for buccal delivery [84]. Chitosan-vancomycin NPs for colon delivery were prepared by two different methods: ion gelation and spray drying [203]. Coco et al. proved the ability of NPs made with chitosan for inflamed colon drug delivery [204]. Several batches of NPs were prepared by entrapping ovalbumin (OVA) into Eudragit S, trimethylchitosan, PLGA, PEG-PLGA and PEG-PCL, separately. Of all the NPs made, NPs with trimethyl chitosan have shown the highest permeability of OVA. However, high permeability was also seen with PEG-PLGA NPs as they were coated with mannose for active targeting of the area of inflammation. As another example, chitosan-carboxymethyl starch nanoparticles of 5-aminosalicylic acid, another drug for inflammatory bowel disease, was prepared, which achieved high entrapment efficiency as well as controlled drug release [205]. Systemic absorption of insulin was demonstrated by the formulation in Ch NPs and administration by the nasal route. Insulin loading up to 55% was achieved and nasal absorption of insulin was greater from Ch NP [206]. Rosmarinic acid-loaded chitosan NPs was prepared by an ion gelation method for ocular delivery. Imiquimod was formulated as chitosan-coated PCL nanocapsules embedded in chitosan hydrogel for vaginal delivery to treat human papillomavirus infection [207]. The pharmacokinetics of chitosan-based formulations has been studied by a few authors [89,208] and a couple of clinical vaccine trials of chitosan formulations have been reported [209,210]. Quinones et al. exhaustively reviewed the chitosan-based self-assembled NPs in drug delivery [211].

Metronidazole (MZ) is an antibiotic with common side-effects of nausea, vomiting, epigastric pain, and mouth dryness, most likely caused by high concentrations of residual MZ in the saliva [212]. To protect MZ from dissolution in saliva, the drug was loaded into Ch NPs of 200–300 nm in size, showing controlled release [213]. Ch NPs loaded with insulin were also developed to improve the systemic delivery of insulin through the nasal passage [214]. The NPs were shown to reduce blood glucose levels by 52.9% in rats and 72.6% in sheep. Ch was chemically modified with the hydrophobic *n*-hexanoic anhydride to form an amphiphilic Ch derivative that showed better blood compatibility [215]. Cefadroxil drug-loaded nanofibers CFX-CPNFs were successfully fabricated by the freeze drying method. The antimicrobial activity indicated that the CFXCPNFs had excellent bacterial activity against 16 strains of Staphylococcus bacteria [216].

Electrospun Ch NFs have been reported for their role in drug delivery [217]. Ch/PVA NFs has been successfully used for enzyme lipase immobilization [218], Ch/ PVA NFs have been used as novel biomedicated nanofibers for preventing wound infections and local chemotherapy [219], Chitosan Carboxymethyl-β-CycloDextrin/PVA has been demonstrated for slow release of CD drug [220], Ch/Polylactic-Co Glycolic Acid (PLGA) showed better fibroblast attachment and proliferation compared with PLGA alone, and Chitosan/phospholipid NFs were attributed to be used as platforms for transdermal drug delivery. The release rate of the model drugs used in this study depended on their solubility [221,222]. Figure 2 gives an overview of the overall drug delivery options successfully demonstrated using Ch NPs.

## 5. Challenges and Future Perspectives

Ch-based nanomaterials are among the most promising polymeric biomaterials being synthesized, because of their distinctive characteristics, biodegradability, non-toxicity, and antimicrobial properties. Ch NPs are extensively studied for biological, biomedical, and pharmaceutical applications, including drug delivery and gene delivery, as well as a therapeutic delivery system and nanosystem for cancer, for wound healing, and as bactericidal agents. Current research is focused on improving the stability, biocompatibility, and synthesis of novel Ch NPs to enhance their effectiveness in biomedical applications [223]. Commonly, Ch was found to be relatively safe due to its biodegradable and biocompatible properties. However, several studies showed the cytotoxicity of Ch NPs in vitro and in vivo. Thus, the present knowledge on Ch-based nanomaterials is not adequate. More research is required to comprehensively investigate the toxicity of Ch NPs for human beings and other living organisms. Moreover, green and environmentally benign synthesis methods for Ch derivatives should be developed to protect the environment [224]. Recently, the concepts of “eco-sustainability”, “reuse”, and “recycling” have become the basis of holistic research approaches.

There are several drawbacks in the use of chitosan for drug delivery systems. The main drawback is its poor solubility at physiological pH owing to the partial protonation of amino groups. To overcome these inherent drawbacks, various derivatives of chitosan, such as carboxylated, different conjugates, thiolated, and acylated chitosan have been devised [225,226]. Researchers reported on the goals of using chitosan as an excipient for drug delivery systems [227,228] and in the development of chitosan drug control releasing systems, including chitosan sponges, chitosan film, chitosan beads, chitosan microbeads (microspheres), and chitosan nanoparticles.

To date, chitosan has shown little or no toxicity in animal models and there have been no reports of major adverse effects in healthy human volunteers; however, clinical data are lacking. Even though chitosan is approved in dietary use, wound dressing applications and cartilage formulations, chitosan-based drug formulation are yet to be approved for mass marketing [228]. Of all the various biomedical applications, we observed that drug delivery applications of Ch NPs are that which are being highly researched and investigated. Ch NPs targeted cancer theranostics, dermatological applications, and targeted parenteral drug delivery systems need to be seriously looked into [229,230,231,232,233,234,235,236]. With the advent of new strategies in overcoming the limitations of chitosan, we expect to see more chitosan research work in near future, especially in nasal and pulmonary drug delivery.

As this review exposes, there are only a few scattered reports on the green synthesis of Ch NPs; this being the latest trend in nanomaterial synthesis, it is rather absurd that not much attention is paid in this direction. This review expects a hike in research publication in the direction of green synthesized Ch NPs, and Ch nanocomposites. In addition, not much work is available on the use of green synthesized Ch NPs for biomedical applications; this is an interesting area worth working on. If a material recovered from marine wastes can be extracted using green methods and put to use for valuable biomedical causes, what are we waiting for?

## 6. Conclusions

The general scenario of chitosan recovery was briefly reviewed in this paper. The scattered reports on green synthesis of chitosan and Ch NPs have been consolidated and presented. The drug delivery milestones by chitosan and its nanoforms has been presented and the lacunae in the application of green synthesized towards drug delivery applications has been highlighted.

## Figures and Tables

**Figure 1 polymers-13-02256-f001:**
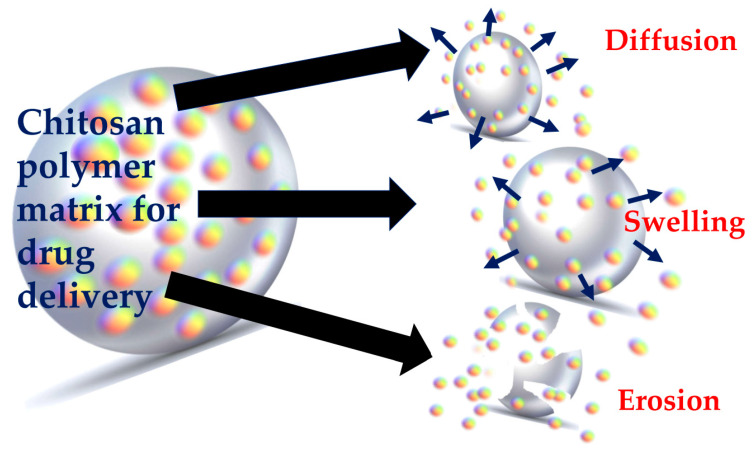
The predominant three processes involved in chitosan polymer-based drug delivery.

**Figure 2 polymers-13-02256-f002:**
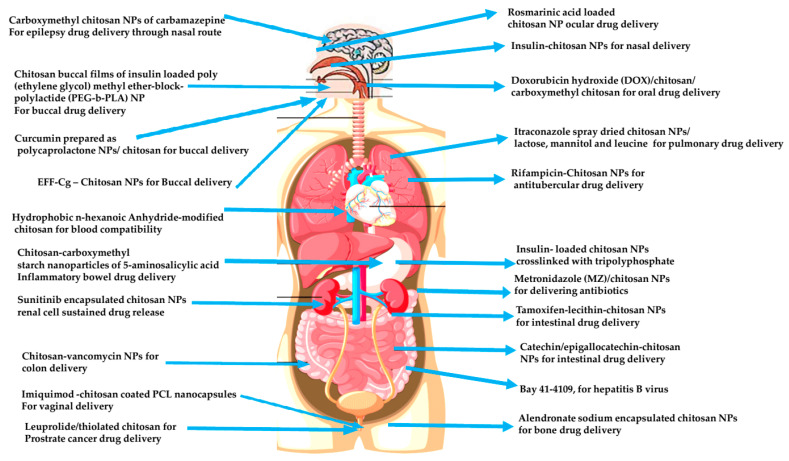
Overview of various drug delivery applications demonstrated by chitosan and its nanoforms.

**Table 1 polymers-13-02256-t001:** List of green synthesized chitin-based materials.

Green Synthesized Chitin Based Material	Chitin/Chitosan/Composites/Chitin NF/Ch Nps	Green Synthesis Method	References
Chitin	Chitin recovered from shrimp shells	NADES, DES and citric acid-based microwave assisted method	[111]
Chitin	Chitin recovered from shrimp shells	NADES-based zero waste method	[115]
Chitosan	Chitin to chitosan	Glycerol-based	[116]
Ch NPs	Ch NPs/ cinnamaldehyde	Crosslinking with ecofriendly cinnamaldehyde	[117]
Chitin NFs	NFs from crab/shrimp/Squid pens	Mechanical grinding	[119,120,121,122]
Chitin NFs	NFs	Using ionic liquids instead of HFIP	[124,125,126,127,128]
Chitosan NFs	Ch NFs	Electrospinning	[130]
Chitosan NFs	Ch NFs	Freeze drying process	[132]
Chitosan NFs	NFs Extracted from crab/shrimp/squid pens	Graviola plant extract mediated recovery	[133]
Ag NPs/Chitosan	Green synthesized Ag NPs entrapped in chitosan	Aqueous extract of tomato plant	[134]
Chitosan based nanocomposite	Silver based chitosan bionanocomposites	synthesized using the stem extract of Saccharum officinarum	[135]
AuNPs/Ch composite	Au NPs in a chitosan matrix	Green route	[136]
Chitosan nanocomposite	Ag/montmorillonite (MMT)/chitosan bionanocomposites	UV irradiation	[137]
Chitosan nanocomposite	Ag/Ch/silica nanocomposites	Electrospinning	[138]
Chitosan nanocomposite	Ag-Ch nanocomposites	Using chitosan	[139]
Chitosan nanocomposite	Au NPs were stabilized into a chitosan matrix	Using chitosan	[140]
Chitosan nanocomposite	Ch-Fe_2_O_3_ nanocomposite	Facile one pot green route	[141]
Chitosan nanocomposite	CA/SA/Ch NPs nanocomposites	Using cinnamaldehyde	[142]
Chitosan nanocomposite	carboxymethyl chitosan-nanosilver	Microwave irradiation	[143]
Chitosan nanocomposite	Ch-modified nano-scale graphene oxide (Ch-NGO) hybrid nanosheets	Ultrasound	[144]
Chitosan nanocomposite	HS/Ch-Nanocomposite	Green route	[145]

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
