# Peer review of "Green Synthesized Chitosan/Chitosan Nanoforms/Nanocomposites for Drug Delivery Applications"

_polymers, 2021, doi:10.3390/polym13142256_

Round 1
Reviewer 1 Report
This review, briefly runs through the various methods involved in the preparation of chitosan and chitosan nanoforms. And mainly focused on the synthesis of chitosan, chitosan nanomaterial and chitosan nanocomposites, where the authors highlighted recent advances in the drug delivery applications of chitosan and its nanoforms. The topic selection of the review is interesting, but there are still some issues in the content organization, and there are too few charts. Hence, some issues should be clearly addressed before it can be accepted. The details are listed as the following:
Introduction:
- Page 2, line 52: “NPs” and “NP” should be “nanoparticles (NPs)” and “nanoparticle (NP)”. And line 56: “nanoparticles (NPs)” should be “NPs”. There are many similar errors, such as page 6, line 301, please check the full text.
- Page 2, line 53 and line 65: the expression of “iron, silver, zinc and titanium oxide” are different, please use the same expression.
- Page 2, line 54: “titanium oxide (TiO2)” should be “titanium oxide (TiO2)”.
Chitosan/chitosan NPs synthesis
- Please separate values from units, e.g., page 3, line 112: “100 °C” not “100°C”; line 118: “3 h” not “3h”; line 122: “65-100 °C” not “65-100°C”.
- Page 3, line 149: “chitosan NPs” should be “Ch NPs”, there are many similar errors, e.g., line 69: “chitosan NPs (Ch NPs)”, while line 265: “chitosan nanoparticles (CSNPs)”. Please check the full text and unify the abbreviations of “chitosan” and “chitosan nanoparticles”.
- The description in 2 Preparation of Ch NPs is the traditional synthesis method of chitosan nanoparticles. Please give examples and lists of the advantages, disadvantages and adaptability of different methods, rather than simple descriptions.
Green Synthesis of Chitin/Chitosan/Ch NPs
- I would suggest adding subheadings such as: “3.1. Green Synthesis of Chitin NPs; 3.2. Green Synthesis of Chitosan NPs; 3.3. Green Synthesis of Chitin/Chitosan/Ch NPs in the third part to make the review clearer.
- Page 4, line 188: please indent “Recently” by 2 characters. And page 5, line 229: please indent “Sini et al.” by 2 characters.
- Page 5, line 221-227: please use the schematic diagram to show the two fermentation methods and compare the two methods.
- Page 5, line 230: what factors do “many factors” refer to? This is unclear and need to be clarified
- Page 5, line 237: “Recently, Younes and Rinaudo have extensively reviewed the preparation methods involved in the recovery of chitin and chitosan from marine sources”, please elaborate further.
- Page 5, line 239-258: three different ways of green synthesis are briefly listed: plant extracts and chitosan co-synthesize chitosan-metal nanoparticles; ultraviolet irradiation to prepare biological nanocomposites and electrospinning to prepare chitosan nanofibers. However, there are many reports on the synthesis of metal nanoparticles using plant extracts, and the use of chitosan as stabilizer and template to synthesize chitosan-metal nanoparticles. I would suggest making a list by referring to some relevant literatures and give your own insights and discussions.
Line 267-277 and 278-280 reported that green synthesis of chitosan-metal nanocomposites using chitosan as a reducing agent as well as a stabilizing agent and NaOH accelerator, which have similarities with the examples in line 243-244. As mentioned earlier, I would suggest to merge and list.
In addition, there are some omissions in the review of chitosan-metal nanoparticles, such as chitosan-zinc oxide nanoparticles and chitosan-CuO nanoparticles.
In general, line 239-341 is disorganized, please classify and describe it.
- Please unify the description of bacteria in Page 5. Line 247; Page 6, line 270 and line 293, such as whether to use abbreviations and italics.
- Page 6, line 278: “CS-Fe2O3” should be “CS-Fe2O3”.
- Page 6, line 281-287: there have been many studies on the preparation of nanoparticles with chitosan and sodium alginate as wall materials. I would suggest explaining the mechanism and further expanding it.
- Page 7, line 326-327: “In the mixing method, polyethylene oxide and poly (vinyl alcohol) (PVA) are often used to blend 327 with chitosan solutions.” References should be added.
- Page 7, line 342-349: whether it is more appropriate to place it in the green synthesis part of chitosan nanoparticles in the front?
Drug Delivery applications of Chitosan/nanochitosan
- In this part, I would also suggest to add subheadings to elaborate to make the review more logical.
- Page 12, line 537: “wound healing”, it seems that the application of chitosan nanoparticles in wound healing does not appear in the article.
- Page 12, line 542-543: “However, several studies showed the cytotoxicity of Ch NPs in vitro and in vivo.” Need to add references.
References
- Please check the format of the references, e.g., page 13, line 596-597: “Adv Drug Deliver Rev” should be “ Drug. Deliver Rev.”
Author Response
This review, briefly runs through the various methods involved in the preparation of chitosan and chitosan nanoforms. And mainly focused on the synthesis of chitosan, chitosan nanomaterial and chitosan nanocomposites, where the authors highlighted recent advances in the drug delivery applications of chitosan and its nanoforms. The topic selection of the review is interesting, but there are still some issues in the content organization, and there are too few charts. Hence, some issues should be clearly addressed before it can be accepted. The details are listed as the following:
Ans. We would like to thank the Editor and Reviewers for the minor revision opportunity. We appreciate your excellent processing speed. We have now revised the manuscript as much as possible, incorporating all your valuable suggestions. We have introduced table 1 in the revision. We have shown the revisions in the text using track changes. We present a point by point response to the queries below. Thank you again.
Introduction:
- Page 2, line 52: “NPs” and “NP” should be “nanoparticles (NPs)” and “nanoparticle (NP)”. And line 56: “nanoparticles (NPs)” should be “NPs”. There are many similar errors, such as page 6, line 301, please check the full text.
Ans. Corrected.
- Page 2, line 53 and line 65: the expression of “iron, silver, zinc and titanium oxide” are different, please use the same expression.
Ans. Corrected.
- Page 2, line 54: “titanium oxide (TiO2)” should be “titanium oxide (TiO2)”.
Ans. Sorry, Corrected. Thank you.
Chitosan/chitosan NPs synthesis
- Please separate values from units, e.g., page 3, line 112: “100 °C” not “100°C”; line 118: “3 h” not “3h”; line 122: “65-100 °C” not “65-100°C”.
Ans. Separated.
- Page 3, line 149: “chitosan NPs” should be “Ch NPs”, there are many similar errors, e.g., line 69: “chitosan NPs (Ch NPs)”, while line 265: “chitosan nanoparticles (CSNPs)”. Please check the full text and unify the abbreviations of “chitosan” and “chitosan nanoparticles”.
Ans. Changed throughout the text.
- The description in 2 Preparation of Ch NPs is the traditional synthesis method of chitosan nanoparticles. Please give examples and lists of the advantages, disadvantages and adaptability of different methods, rather than simple descriptions.
Ans. The traditional synthesis of Ch NPs is already well represented and elaborated in multiple reviews, we have restricted ourselves to a brief representation. We have now added citations of the reviews that cover this topic in the revised manuscript in Section 2. Thank you.
Green Synthesis of Chitin/Chitosan/Ch NPs
- I would suggest adding subheadings such as: “3.1. Green Synthesis of Chitin NPs; 3.2. Green Synthesis of Chitosan NPs; 3.3. Green Synthesis of Chitin/Chitosan/Ch NPs in the third part to make the review clearer.
Ans. Yes, we absolutely agree. We thank you for the valuable suggestions, we have now added subheading, there aren’t much under chitin and chitosan green synthesis to order under subheadings, hence we have partitioned the section on a different vertical. But, this reorientation has made the section clearer. Thank you.
- Page 4, line 188: please indent “Recently” by 2 characters. And page 5, line 229: please indent “Sini et al.” by 2 characters.
Ans. Indented
- Page 5, line 221-227: please use the schematic diagram to show the two fermentation methods and compare the two methods.
Ans. We think this is not exactly the focus of this review. Many reviews have described these methods in depth and detail. Thank you.
- Page 5, line 230: what factors do “many factors” refer to? This is unclear and need to be clarified
Ans. removed
- Page 5, line 237: “Recently, Younes and Rinaudo have extensively reviewed the preparation methods involved in the recovery of chitin and chitosan from marine sources”, please elaborate further.
Ans. Elaborated slightly.
- Page 5, line 239-258: three different ways of green synthesis are briefly listed: plant extracts and chitosan co-synthesize chitosan-metal nanoparticles; ultraviolet irradiation to prepare biological nanocomposites and electrospinning to prepare chitosan nanofibers. However, there are many reports on the synthesis of metal nanoparticles using plant extracts, and the use of chitosan as stabilizer and template to synthesize chitosan-metal nanoparticles. I would suggest making a list by referring to some relevant literatures and give your own insights and discussions.
Ans. Yes, we do understand your concern, however, we are choosing to go with focusing on green synthesis of chitosan and NPs rather than the use of chitosan in metal NP synthesis. We have added a sentence on this in the revision as per your suggestion. But elaborating on this would be beyond the actual point of this review. Thank you for your understanding.
Line 267-277 and 278-280 reported that green synthesis of chitosan-metal nanocomposites using chitosan as a reducing agent as well as a stabilizing agent and NaOH accelerator, which have similarities with the examples in line 243-244. As mentioned earlier, I would suggest to merge and list.
Ans. Merged
In addition, there are some omissions in the review of chitosan-metal nanoparticles, such as chitosan-zinc oxide nanoparticles and chitosan-CuO nanoparticles.
Ans. Added
In general, line 239-341 is disorganized, please classify and describe it.
Ans. Organized and rewritten.
- Please unify the description of bacteria in Page 5. Line 247; Page 6, line 270 and line 293, such as whether to use abbreviations and italics.
Ans. Unified
- Page 6, line 278: “CS-Fe2O3” should be “CS-Fe2O3”.
Ans. Corrected
- Page 6, line 281-287: there have been many studies on the preparation of nanoparticles with chitosan and sodium alginate as wall materials. I would suggest explaining the mechanism and further expanding it.
Ans. Again this would be a deviation, we restrict to chitosan preparation and green synthesis and not chitosan use in green synthesis of other NPs. Thank you.
- Page 7, line 326-327: “In the mixing method, polyethylene oxide and poly (vinyl alcohol) (PVA) are often used to blend 327 with chitosan solutions.” References should be added.
Ans. Added.
- Page 7, line 342-349: whether it is more appropriate to place it in the green synthesis part of chitosan nanoparticles in the front?
Ans. We have rewritten this section; it was a mess. Now we have organized it to have a flow.
Drug Delivery applications of Chitosan/nanochitosan
- In this part, I would also suggest to add subheadings to elaborate to make the review more logical.
Ans. Added
- Page 12, line 537: “wound healing”, it seems that the application of chitosan nanoparticles in wound healing does not appear in the article.
Ans. Corrected
- Page 12, line 542-543: “However, several studies showed the cytotoxicity of Ch NPs in vitro and in vivo.” Need to add references.
Ans. Added
References
- Please check the format of the references, e.g., page 13, line 596-597: “Adv Drug Deliver Rev” should be “ Drug. Deliver Rev.”
Ans. Corrected.
Reviewer 2 Report
The review by I. Sivanesan et al. is focused on green synthesis of chitosan and its various forms, including nanocomposites, with particular emphasis on drug delivery applications. The manuscript includes three main sections; the former dedicated to the conventional techniques to obtain chitosan and ch-NPs, the second which deals with green ways to synthesize chitin, chitosan and ch-NPs and the latter, focused on various applications in drug delivery systems. I personally find the section dedicated to green synthesis very interesting; the use of enzymes and microorganisms, once optimized, would allow at the same time to preserve the chitin structure and to reduce the usage of chemicals, reducing thus e.g. their presence in the wastewater effluents deriving from extraction.
In general, the review is well-written, well-organized and properly referenced; it contains an exhaustive description of drug delivery applications of chitosan and nanocomposites chitosan-based although it can be slightly improved in some points. Minor recommendations to improve the quality of the paper are listed below:
-In the section dedicated to the preparation of chitosan nanoparticles, for completeness the authors could refer also to the relevant case of self-assembly approach, which can be obtained e.g. by grafting hydrophobic moieties within the chitosan structure and/or through the formation of polyelectrolyte complexes with polyanions. You may critically use the following paper (https://doi.org/10.3390/polym10030235).
-at lines 478-480, authors state ‘…chitosan exhibits antibacterial activity by binding to phosphoryl groups and lipopolysaccharides on bacterial cell membranes which benefits in fighting pulmonary bacterial infections’. Did the authors refer to a specific study? If yes, please add it. Did the antibacterial activity mentioned has been proved on live cell or model systems? Especially in the latter case, lipid composition as well as the presence of pulmonary surfactant has to be carefully taken into account.
-at lines 542-543, authors state ‘However, several studies showed the cytotoxicity of Ch NPs in vitro and in vivo’. For greater clarity, some general considerations about the intrinsic cytotoxicity of Ch NPs should be addressed earlier in the text. Adding some considerations on how the use of green chemistry can regulate such cytotoxic effects would increase the readability of the review.
-the authors often refer to composites materials chitosan-based: incorporation/functionalization of metallic nanoparticles is obtained with several methods including the reduction of salt precursor. In this context a brief mention to these aspects, including the particular case which allow to control size and shape of metallic nanostructures embedded in chitosan matrix could help the reader having a more complete vision. You can use the following papers (http://dx.doi.org/10.1021/am508094e; https://doi.org/10.1088/1361-6528/aa8337; https://doi.org/10.1002/adma.201002228;)
-Minor typos and punctuation needs to be checked. Below, the list of some found in the text:
Line 412: of chitosan instead of ‘od chitosan’
Line 458: sustained, S capital
Line 484: solubility. instead of solubility,
Line 507: by formulationin
Author Response
The review by I. Sivanesan et al. is focused on green synthesis of chitosan and its various forms, including nanocomposites, with particular emphasis on drug delivery applications. The manuscript includes three main sections; the former dedicated to the conventional techniques to obtain chitosan and ch-NPs, the second which deals with green ways to synthesize chitin, chitosan and ch-NPs and the latter, focused on various applications in drug delivery systems. I personally find the section dedicated to green synthesis very interesting; the use of enzymes and microorganisms, once optimized, would allow at the same time to preserve the chitin structure and to reduce the usage of chemicals, reducing thus e.g. their presence in the wastewater effluents deriving from extraction.
Ans. We would like to thank the Editor and Reviewers for the minor revision opportunity. We thank you for your encouraging comments and nice words. We appreciate your excellent processing speed. We have now revised the manuscript as much as possible, incorporating all your valuable suggestions. We have shown the revisions in the text using track changes. We present a point by point response to the queries below. Thank you again.
In general, the review is well-written, well-organized and properly referenced; it contains an exhaustive description of drug delivery applications of chitosan and nanocomposites chitosan-based although it can be slightly improved in some points. Minor recommendations to improve the quality of the paper are listed below:
Ans. Thank you again for your recommendation. We have improved and organized the drug delivery section.
-In the section dedicated to the preparation of chitosan nanoparticles, for completeness the authors could refer also to the relevant case of self-assembly approach, which can be obtained e.g. by grafting hydrophobic moieties within the chitosan structure and/or through the formation of polyelectrolyte complexes with polyanions. You may critically use the following paper (https://doi.org/10.3390/polym10030235).
Ans. Added. Thank you.
-at lines 478-480, authors state ‘…chitosan exhibits antibacterial activity by binding to phosphoryl groups and lipopolysaccharides on bacterial cell membranes which benefits in fighting pulmonary bacterial infections’. Did the authors refer to a specific study? If yes, please add it. Did the antibacterial activity mentioned has been proved on live cell or model systems? Especially in the latter case, lipid composition as well as the presence of pulmonary surfactant has to be carefully taken into account.
Ans. Added. And checked.
-at lines 542-543, authors state ‘However, several studies showed the cytotoxicity of Ch NPs in vitro and in vivo’. For greater clarity, some general considerations about the intrinsic cytotoxicity of Ch NPs should be addressed earlier in the text. Adding some considerations on how the use of green chemistry can regulate such cytotoxic effects would increase the readability of the review.
Ans. Agreed, addressed as per your suggestion. Thank you.
-the authors often refer to composites materials chitosan-based: incorporation/functionalization of metallic nanoparticles is obtained with several methods including the reduction of salt precursor. In this context a brief mention to these aspects, including the particular case which allow to control size and shape of metallic nanostructures embedded in chitosan matrix could help the reader having a more complete vision. You can use the following papers (http://dx.doi.org/10.1021/am508094e; https://doi.org/10.1088/1361-6528/aa8337; https://doi.org/10.1002/adma.201002228;)
Ans. We have already exceeded the word limit, hence we have restricted from extending into this area. Thank you for your kind understanding.
-Minor typos and punctuation needs to be checked. Below, the list of some found in the text:
Line 412: of chitosan instead of ‘od chitosan’
Ans. Corrected. Sorry about that.
Line 458: sustained, S capital
Ans. Corrected.
Line 484: solubility. instead of solubility,
Ans. Corrected.
Line 507: by formulationin
Ans. Corrected. Sorry about that.